# Multi-Trait Wheat Rhizobacteria from Calcareous Soil with Biocontrol Activity Promote Plant Growth and Mitigate Salinity Stress

**DOI:** 10.3390/microorganisms9081588

**Published:** 2021-07-26

**Authors:** Anastasia Venieraki, Styliani N. Chorianopoulou, Panagiotis Katinakis, Dimitris L. Bouranis

**Affiliations:** 1Laboratory of Plant Pathology, Crop Science Department, Agricultural University of Athens, Iera Odos 75, 118 55 Athens, Greece; 2Laboratory of Plant Physiology and Morphology, Crop Science Department, Agricultural University of Athens, Iera Odos 75, 118 55 Athens, Greece; s.chorianopoulou@aua.gr (S.N.C.); bouranis@aua.gr (D.L.B.); 3Laboratory of General and Agricultural Microbiology, Crop Science Department, Agricultural University of Athens, Iera Odos 75, 118 55 Athens, Greece; katp@aua.gr

**Keywords:** biofertilizers, arylsulfatase (ARS)-producing bacteria, plant growth promoting rhizobacteria (PGPR), plant growth promoting traits, salinity tolerance, bacterial strain compatibility

## Abstract

Plant growth promoting rhizobacteria (PGPR) can be functional microbial fertilizers and/or biological control agents, contributing to an eco-spirit and safe solution for chemical replacement. Therefore, we have isolated rhizospheric arylsulfatase (ARS)-producing bacteria, belonging to *Pseudomonas* and *Bacillus* genus, from durum wheat crop grown on calcareous soil. These isolates harbouring plant growth promoting (PGP) traits were further evaluated in vitro for additional PGP traits, including indole compounds production and biocontrol activity against phytopathogens, limiting the group of multi-trait strains to eight. The selected bacterial strains were further evaluated for PGP attributes associated with biofilm formation, compatibility, salt tolerance ability and effect on plant growth. In vitro studies demonstrated that the multi-trait isolates, *Bacillus* (1.SG.7, 5.SG.3) and *Pseudomonas* (2.SG.20, 2.C.19) strains, enhanced the lateral roots abundance and shoots biomass, mitigated salinity stress, suggesting the utility of beneficial ARS-producing bacteria as potential microbial fertilizers. Furthermore, in vitro studies demonstrated that compatible combinations of multi-trait isolates, *Bacillus* sp. 1.SG.7 in a mixture coupled with 5.SG.3, and 2.C.19 with 5.SG.3 belonging to *Bacillus* and *Pseudomonas*, respectively, may enhance plant growth as compared to single inoculants.

## 1. Introduction

Root beneficial microbiome plays a crucial and significant role in sustainable agriculture. A free-living natural root microbiome, applied as microbial fertilizer and/or as a biological control agent, interacts with the host conferring an eco-friendly balance upon the plant-microbe-soil system. These rhizospheric interactions establish the initial and crucial conditions to improve soil fertility and plant health. Root microbes and secretions from both roots and microbes influence the rhizosphere, the plant growth stages and its resistance against phytopathogens and abiotic stresses [1,2,3]. The interaction of plant roots with the rhizosphere microbiome constitutes the plant–root microbiome [4]. Free-living soil bacteria that colonize plant roots and affect plant growth, referred to as plant growth promoting rhizobacteria (PGPR). Many PGPR strains have been used as biofertilizers, successfully contributing to sustainable agriculture [5,6]. Biofertilizers do not cause harmful effects to the ecosystem after a prolonged period in the soil, thus making their use in agriculture more enticing [7,8,9,10]. However, not all PGPR are suitable as biofertilizers. Many PGPR promote plant growth and can be biocontrol agents at the same time, but if rhizospheric bacterial strains are characterized by biocontrol attributes only, they are simply biocontrol agents [11]. Βiofertilising-PGPR can consist of a single strain or a consortium. In both cases, they display several modes of action [11,12]. The combination of the plethora of modes of actions of each strain, combined to their possible compatibility between different strains can result in multi-functional microbial fertilizer. Consortia of PGPR or other beneficial microorganisms, in many instances, are more effective compared to single strain. The compatibility issue among the strains of which the consortium consists is not new. It has been discussed since the beginning of the PGPR research, and there is an increasing necessity to create sustainable and effective multistrain mixtures [12,13,14,15,16]. A compatible mixture in the field ensures that the advantages of the bacterial strains are long-lasting and do not neutralize each other. Molina-Romero et al., 2017 [17], have proved that a compatible bacterial mixture of four strains (*Pseudomonas putida* KT2440, *Sphingomonas* sp. OF178, *Azospirillum brasilense* Sp7, *Acinetobacter* sp. EMM02) were able to adhere to seeds and colonize the rhizosphere of plants when applied in both mono-inoculation and multi-inoculation treatments, showing that they can also coexist without antagonistic effects in association with plants.

Bacterial, as well as fungal microorganisms produce arylsulfatases towards sulphate ester hydrolysis of the soil organic matter to uptake the released inorganic sulphate [18,19]. Bacterial arylsulphatase enzymes have been studied in medicine, industry and food science. Researchers managed to improve the thermostability of *Pseudoalteromonas carrageenovora* arylshulphatase developing mutants [20] and its enzyme activity by directed evolution followed by characterizing the biochemical properties of the selected mutant enzyme, making a promising tool in industrial applications for agar quality improvement [21]. Arylsulfatases of some bacterial species have been already characterized as *Klebsiella pneumoniae* [22], *Kluyveromyces lactis* [23], *Pseudoalteromonas carrageenovora* [24], *Pseudomonas aeruginosa* [25], *Salmonella typhimurium* [26], *Serratia marcescens* [10,27] and *Thermotoga maritima* [28]. Arylsulfatase plays an important role in cancer detection [29] and contributes to the doping analysis and food processing [19]. Moreover, it improves the quality of agar in industry [20], increases the inorganic sulphate of soil [30] and indicates healthy plant cultivation and crop rotations [31]. However, limited systematic studies have been performed concerning ARS-producing microorganisms in agricultural soils.

It is known that, when preferred S sources become limiting, bacteria synthesize certain proteins to overcome elemental deficiencies, named sulphate starvation induced (SSI) proteins. They are implicated in different strategies that the cell can develop to satisfy its S requirement. Arylsulfatase enzymes (ARS) are a group of the SSI proteins produced by microorganisms during S starvation. ARS have been key enzymes in the bacterial cell response against S limitation, and they can potentially hydrolyze sulphate esters from environmental sources or from the reserves of the bacterial cell [18,19]. In agriculture, arylsulfatase is one of the enzyme activities used to evaluate the soil biological diversity in natural or burdened ecosystems due to intensive crop systems, application of pesticides, and overdosed fertilization [19,31,32,33,34,35,36,37]. Increased arylsulfatase activity in soils is an indication of plant cultivation and crop rotations [31,32,33,34] and vice versa; the decrease of arylsulfatase activity in field soil is an indication of metal and pesticides contaminations [33,34,35,36]. A soil with arylsulphatase microbial activity improves or alters its biological composition by affecting its fertility [35,36,37,38]. In addition to arylsulphatase microbial activity, there are other enzymatic microbial activities such as dehydrogenases and β-glucosidase, which are indications of good soil quality [19,36,37,38,39].

The presence of excess amounts of salts in agricultural soil is one of the common and major abiotic stresses that reduces crop growth and yield, thus concerning many people worldwide. The problem grows and swells over time because the salinized soil area increases at a rate of 10% per year [40]. Therefore, maintaining the ecological balance of the agricultural ecosystems requires the imposition of drastic solutions to the problem [3,40]. Many studies prove the effect of PGPR on the alleviation of salt stress on crop plants [13,41,42,43,44,45]. *Arthrobacter protophormiae* (SA3) and *Dietzia natronolimnaea* (STR1) enhanced photosynthetic efficiency [42], IAA content and other beneficial traits in wheat, *Klebsiella*, *Pseudomonas*, *Agrobacterium*, and *Ochrobactrum* in peanut [43], *Pseudomonas putida* UW4 and *P. migulae* 8R6 effected on *Camelina sativa* (camelina) [44], *Micrococcus yunnanensis*, *Planococcus rifietoensis*, and *Variovorax paradoxus* on *Beta vulgaris* [45], *Bacillus licheniformis* AP6 and *Pseudomonas plecoglossicida* PB5 with proven biofilm formation can facilitate sunflower growth by improving physiological and biochemical attributes [46], *Bacillus megaterium* in *Zea mays*, *Azospirillum* sp. in lettuce, *Achromobacter piechaudii* in *tomato*, *Eneterobacter* sp. PR14 in rice and millets [13], whilst two *Bacillus* spp. strains mixed with nanozeolite were applied to the seeds on maize crop [47] and other.

We demonstrated that the field application of fertilizer granules with incorporated elemental sulphur (FBS^0^) affected the abundance of wheat rhizospheric arylsulfatase (ARS)-producing bacteria communities and effectively enhance the microbially mediated nutrient mobilization towards improved plant nutritional dynamics [48].

In this report, our aims are: (i) further screening the arylsulfatase (ARS)-producing bacteria for beneficial traits, (ii) assessing the growth promotion activity of the selected rhizospheric arylsulfatase (ARS)-producing bacteria on a model plant, testing their ability to form biofilms and evaluate their resistance under salinity stress in vitro, (iii) evaluating the effectiveness of selected salinity tolerant and compatible bacterial isolates on plant growth promotion under salinity stress in vitro and (iv) evaluating compatible mixtures of these multi-trait isolates for their plant growth promotion ability.

Our results showed that a selected group of arylsulfatase (ARS)-producing bacterial isolates with biocontrol activity and resistance to salinity stress can stimulate seed germination and enhance plant growth under salinity stress conditions, and also, in vitro studies demonstrated that compatible mixtures of these multi-trait isolates could promote plant growth as compared to single strains.

## 2. Materials and Methods

### 2.1. Bacterial Strains

Sixty-eight arylsulfatase (ARS)-producing bacterial isolates used in this study were isolated from the rhizosphere of durum wheat on calcareous soil after treatment with fertilizer granules containing elemental sulphur (S^0^) and characterized with respect to their phylogenetic affiliation as well as their traits associated with plant nutrition [48]. Briefly, the rhizospheric bacterial strains were isolated from rhisopshaeric soil at different sampling time points through the cultivation period. Cultivable bacterial isolates were collected from modified M9 minimal medium supplemented with the chromogenic arylsulfatase substrate 5-bromo-4-chloro-3-indolyl sulphate (X-Sulf, Sigma-Aldrich®, Merk KGaA, Burlington, MA, USA) (100 mg L^−1^) as the sole sulphur source [49]. Microbial colonies possessing arylsulphatase activity were detected by their blue colour on Petri dishes. Apart from the arylsulfatase activity of the selected strains, we investigated other possible PGP traits, such as phosphate solubilization on Pikovskaya agar [50] and siderophore production on chrome azurol-S (Sigma-Aldrich®, Merk KGaA, Burlington, MA, USA) agar medium [51], and urease production ability was tested by inoculating Urea Base Christensen ISO 6579, ISO 19250 (Conda, Madrid, Spain) medium [48]. All assays were done in triplicate. We selected 68 isolates with the most promising results in beneficial trait analysis, and we identified them by using PCR amplification 16S rRNA gene sequence with the primer set fD1 (5′-AGAGTTTGATCCTGGCTCAG-3′) and rP2 (5′-ACGGCTACCTTGTTACGACTT-3′), targeting ribosomal DNA of approximately 1500 bp. Phylogenetic analysis showed us that most of the isolates belonged to the *Pseudomonas* and *Bacillus* genus and were classified among the clades of beneficial *Pseudomonas*, *Bacillus*, and other nonphytopathogenic antagonistic strains deposited in public databases [48] The nucleotide sequence data are available at the European Nucleotide Archive GenBank database under accession numbers LR027392 to LR027460, BioProject: PRJEB28499. In this study, we used the aforementioned strains for further evaluation by limiting the most promising group strain in each screening step and focusing on selected multi-trait strains in order to evaluate them as potential biocontrol biostimulators/biofertilizers.

### 2.2. Experimental Strategy

The initial selection screening was based on their remarkable antifungal activity against phytopathogenic fungi, as well as on their Indole Acetic Acid (IAA) production. We continued with testing the biofilm associated traits to ensure that these strains are functional colonizers not only in normal but also under arid conditions. Then, we applied a second selection screening among the selected ones after the antifungal and IAA assays, based on the plant growth promoting features. This last group of strains that came up after the second selection was tested for their survival and effectiveness under salinity conditions. All the strains from the two selection steps were tested for their compatibility in dyads, towards using them as biostimulators in consortia. Finally, we selected from the last four strains two compatible consortia (dyads) to test for their growth promoting ability comparing to the single strains (Figure 1).

### 2.3. Antifungal Activity Against Phytopathogenic Fungi

In our previous study, we focused on antifungal properties of the arylsulfatase producing strains on swarming (0.5% agar) conditions [48]. In this study, we extended our biocontrol test experiments on non-swarming (1.5% (*w*/*v*) agar) conditions in all the arylsulphatase producing strains, focussing on the strains that showed remarkable antifungal activity. The antifungal activity was tested against fungi from the initial screening such as *Rhizoctonia solani* and *Fusarium oxysporum* and, additionally, against the phytopathogenic fungi *Botrytis cinerea* and *Colletotrihum* sp. in dual culture in Nutrient Agar dishes incubated for 7–10 days at 25 °C as described in Bouranis at al., 2019 [48]. Determination of beneficial strain antifungal inhibition rate (IR) percentage was calculated using the following formula with respect to positive control.

IR (%) = 100 (X − Y)/(X)
X: diameter (mm) of control mycelium, Y: diameter (mm) of the mycelium in the presence of the beneficial strain.

All assays were done in triplicate.

### 2.4. IAA Production

Isolates were evaluated for their ability to synthesize IAA, thus stimulating plant growth. IAA production was determined by using the Salkowski reagent method. IAA production in the medium leads to a red-pink colour formation). In brief, 10 mL of LB medium supplemented with 0.1% (*w*/*v*) L-tryptophan (Sigma-Aldrich®, Merk KGaA, Burlington, MA, USA) were inoculated with fresh bacterial cultures and incubated under shaking for 72 h at 28 °C. Then, 1 mL of supernatant, obtained by centrifugation (5000× *g*/10 min at 4 °C), was mixed with 3 mL Salkowski’s reagent (2% 0.5 M FeCl_3_ in 35% HCLO_4_ solution) and kept in the dark. After 30 min incubation, IAA amount was spectrophotometrically determined (540 nm) [52,53].

### 2.5. Selection of 8 Bacterial Strains Based on Their Best Trait Analysis

At this time point of the experiment, we selected eight strains out of the 68 ones with the best trait analysis, namely, 1.SG.7 identified as *Bacillus* sp. after 16S rRNA analysis, 2.SG.20 as *P. koreensis*, 5.SG.3 as *B. amyloliquefaciens*, 2.C.19 as *P. moraviensis*, 3.SG.19 and 2.C.23 as *P. fluorescens*, 4.SG.6 and 2.SG.8 as *P. koreensis*, respectively.

### 2.6. Biofilm Associated Traits

#### 2.6.1. Swarming Motility and Temperature Tolerance

All eight bacterial strains (*Bacillus* sp. 1.SG.7, *P. koreensis* strain 2.SG.8, 4.SG.6 and 2.SG.8, *B. amyloliquefaciens* 5.SG.3, *P. moraviensis* 2.C.19, *P. fluorescens* 3.SG.19 and 2.C.23) were grown in *swarm plates* 0.5% (*w*/*v*) Nutrient Agar by *spotting* with 5 µL of overnight liquid broth *culture* and incubated at 30, 37 and 42 °C for 48 h. Control experiment was performed in 1.5% (*w*/*v*) Nutrient Agar *plates*.

#### 2.6.2. Biofilm Formation

The assay for biofilm formation was performed on microtitre plates using crystal violet [54]. The bacterial cells were grown to OD_600_ = 2.0 in Nutrient Broth (NB) (Sigma-Aldrich®, Merk KGaA, Burlington, MA, USA) medium. Then, they were washed and resuspended in the same medium to OD_600_ = 0.2. 200 μL of cells and NB alone for control were added to individual wells of a 96-well PVC plate. The plates were sealed with sterile rayon adhesive film and incubated at 30 °C. At the final time point, the medium was removed, and the biofilms were stained with 200 μL 0.01% crystal violet for 20 min. Excess dye was washed away with five changes of sterile distilled water. The dye was then solubilized with 200 μL (*v*/*v*) 95% ethanol to destain the wells, and the amount of dye was quantified by measuring the absorbance at 570 nm.

### 2.7. Salinity Tolerance of Bacterial Strains

To further investigate the beneficial characteristics under different salt stress, the 8 strains were tested for the ability to grow at different NaCl concentrations (0, 0.2 M, 0.5 M and 1 M). Serial dilutions were prepared up to 10^−7^ CFU/mL in a sterile tube by adding 0.1 mL aliquots from each dilution. A volume of 0.005 mL of aliquot was taken from each dilution and spotted on Petri plate containing Nutrient agar supplemented with the appropriate NaCl concentration. After spotting, the plates were incubated for 48 h at 30 °C temperature.

### 2.8. Strain Compatibility

A compatibility assay was performed among the 8 potential PGP strains using overlay method and drop techniques [55]. Briefly, in the drop technique, we incubated the bacterial strains for 48 h at 30 °C, 120 rpm. After sufficient growth, 5 μL of 1.10^8^ CFU/mL of each isolate were placed on agar plates containing 50 μL of an overnight culture of indicator bacteria spread as a lawn, followed by incubation for 24 h at the same with the indicator conditions. Compatibility was detected by forming an inhibition zone where the drop was placed and recorded qualitatively, based on transparency degree. All eight strains were examined for compatibility with all the possible combinations in dyads between them (64 dyads). Each experiment was performed in triplicate.

### 2.9. Screening of the 8 Selected Bacterial Strains for Plant Growth Promotion on Arabidopsis thaliana (Col-0)

Surface sterilized *A. thaliana* Col-0 seeds with 5% sodium hypochlorite (*v*/*v*) supplemented with 0.1% Triton × 100 (*v*/*v*) for 5 min and washed five times with sterile distilled water were germinated on half-strength Murashige and Skoog (MS ½) medium, including vitamins (MS0222, Duchefa Biochemie, Haarlem, The Netherlands). After stratification at 4 °C for 2–3 days, seeds were sown on MS (½) solid medium supplemented with 0.5 % (*w*/*v*) sucrose. Plates were kept vertically in the growth chamber at 22 °C with a photoperiod of 16 h light and 8 h dark. Four days after germination, uniform seedlings of similar size were transferred onto new (MS ½) solid medium plates, ten seedlings per dish, as described previously.

Bacterial strains were maintained on Nutrient Broth (NB) medium. A colony was inoculated in NB and grown overnight at 180 rpm, 28 °C. The next day, the bacterial culture was diluted to 10^8^ CFU/mL and separated in aliquots of usually 180 µL for each plate [56]. Bacterial strains were then inoculated at a distance of 3 cm from the root tip, and plates were placed vertically in a growth chamber for 15 days. After 15 days, the morphological characteristics such as dry and fresh weight and primary and secondary root length of each whole plant were determined. Observations were done by stereo microscope Leica Wild M3B. Measures were made by the ImageJ software [57]. The experiments were repeated three times.

### 2.10. Screening of the 4 Selected Bacterial Strains for Plant Growth Promotion on A. thaliana (Col-0) under Salinity Conditions

To investigate the growth-promoting ability of the strains under salinity conditions, we selected the four most salt tolerant strains of the arylsulphatase bacterial strains under different salt stress conditions, i.e., we tested the response of *A. thaliana* Col-0 explants in presence of bacterial strain inoculum at different NaCl concentrations (0, 37.5, 75, 150 and 200 mM) under in vitro conditions. Experimental strategies for *A. thaliana* Col-0 seeds and bacterial strain inocula were conducted in the same way as described above. Explants’ survival on these salinity conditions and morphological characteristics such as dry and fresh weight and root length of each plant were recorded.

### 2.11. Screening of 2 Selected Compatible Bacterial Strain Dyads for Plant Growth Promotion on A. thaliana (Col-0)

To investigate the growth-promoting ability of compatible bacterial strain mixtures, 2 compatible dyads were tested, the bacterial mixture Dyad A (2.C.19 coupled with 5.SG.3) identified as *P. moraviensis* and *B. amyloliquefaciens*, respectively, and Dyad B (1.SG.7 coupled with 5.SG.3) identified as *Bacillus* sp. and *B. amyloliquefaciens*, respectively. To determine their positive plant growth effect, the two dyad mixtures were co-cultivated with *A. thaliana* seedlings, inoculated at a distance. Experimental strategy for *A. thaliana* Col-0 seeds was conducted the same way as described above. The two different bacterial strain liquid inocula for each mixture were conducted the same way as described above separately for each strain. They were diluted to 10^8^ CFU/mL and mixed together prior to the inoculation of *Arabidopsis* in vitro. Morphological characteristics such as dry and fresh weight, root length of each plant and lateral root number were recorded.

### 2.12. Statistical Analysis

The comparisons between the result values in each case were performed using one-way ANOVA and Tukey’s honest significant difference post hoc test at *p* < 0.05.

## 3. Results

### 3.1. Antifungal Activity Against Phytopathogenic Fungi

Strains were tested for their antifungal activity against *Rhizoctonia solani* and *Fusarium oxysporum*, as well as against *Botrytis cinerea* and *Colletotrichum* sp. Eight strains showed remarkable inhibition zone against the aforementioned phytopathogens (Figure 2).

The *B. cinerea* inhibition rate in the presence of these strains was up to 54–63% for all these beneficial isolates, except for the presence of *Pseudomonas korrensis* 2SG20 which was 50%. The *F. oxysporum* inhibition rate was up to 53–70% for all the beneficial isolates, and *P. korrensis* 2SG20 was 37% (dual culture in NA dishes, 7 days). Antifungal activity of the strains against *Rhizoctonia solani* and *Colletotrichum* sp. were similar, meaning that inhibition rate of all 8 strains for both aforementioned pathogens was about 55%–65% (Table 1).

### 3.2. IAA Production

Strains 2SG20, 4SG6, 5SG3, 2C19 and 2C23 produced a remarkable amount of IAA expressed by the colour change. Qualitative IAA production revealed that the aforementioned five isolates produced IAA, whilst three shown negative results (1SG7, 2SG8, 3SG9). Quantitative IAA production confirmed the qualitative ones. A varying level of IAA production was recorded for the lower production of 2.35 ± 0.6 μg/mL for isolate 1SG7 to the highest production of 28.35 ± 0.8 and 25.15 ± 0.5 μg/mL for isolates 5.SG.3 and 2.C.19, respectively, as well as the strains 2.C.23, 4.SG.6 and 3.SG.19 with 22.61 ± 0.8, 20.16 ± 0.9 and 18.41 ± 0.2 μg/mL, respectively. Strains 2.SG.8 and 2.SG.20 presented a medium IAA production of 11.2 ± 0.9 and 16.25 ± 0.9 μg/mL, however, also remarkable compared to the mock sample 0.68 ± 0.08 (Table 1).

### 3.3. First Bacterial Strain Selection

Bacterial strains possessing the multi plant growth promoting traits after their further screening for their antagonistic activity against *Rhizoctonia solani*, *Fusarium oxysporum*, *Botrytis cinerea* and *Colletotrihum* sp. in dual culture (a strong antagonistic activity is indicated by the width of inhibition zone) and their production of indole related compounds (an indicator of indole acetic acid production) were selected for further evaluation as potential biocontrol biofertilizers. Eight out of the 68 bacterial strains (hereafter referred to as multi-trait bacterial strains) showed strong antagonistic activity against fungi forming an inhibition zone with a width larger than 3 mm) and detectable levels of indole compounds production and selected for further studies (Table 1).

### 3.4. Further Beneficial Traits

The selected bacterial strains possessed traits associated with biofilm formation, tolerance to salinity and temperature, root colonization and community establishment (swarming ability, biofilm formation and compatibility).

#### 3.4.1. Biofilm Associated Traits

##### Swarming Motility and Temperature Tolerance

All 8 strains were grown in swarming conditions (0.5% NA) at 30, 37 and 42 °C for 48 h. The strains *P. fluorescens* 2C23, *Paenibacillus polymyxa* 5SG10, *Bacillus amyloliquefaciens* 5SG3 managed to grow a swarming colony at 42 °C. Such strains with high swarming motility and temperature tolerance are remarkable root colonizers.

##### Biofilm Formation

Most of our multi-trait isolates showed strong biofilm development. The isolate 1SG7 showed strong biofilm development with optical density 1.12 ± 0.095 at 590 nm, which was an impressive one (Appendix A).

#### 3.4.2. Salinity Tolerance

##### Salt Tolerance of Bacterial Strains

All strains were salt tolerant up to 0.5 M NaCl. Surprisingly, *Bacillus* sp. strain 1SG7 and *Bacillus amyloliquefaciens* 5SG3 were tolerant isolates and regularly grew up to 1 M NaCl, up to 1.10^8^ CFU/mL and 6.10^7^ CFU/mL ml, respectively (Appendix A).

#### 3.4.3. Strain Compatibility

Compatibility was detected by the formation of an inhibition zone where the drop was placed after at least 24 h, based on their colony growth or their transparency (Figure 3).

We examined all eight strains with all the possible combinations in dyads between them, creating 64 dyads, and we selected the compatible ones (Table 2). Most of the strains were found to be compatible except for (i) 2.C.23 in dyad with 1.SG.7, 2.SG.20, 5.SG.3; (ii) 2.SG.8 with 2.SG.19 and (iii) 2.C.19 in dyad with 2.SG.8 (Figure 3, Table 2).

#### 3.4.4. Plant Growth Promotion on *A. thaliana* (Col-0)

To determine their effect on plant growth, the multi-trait bacterial strains were co-cultivated with *A. thaliana* seedlings (Figure 4). Results revealed that all bacterial strains enhanced the number of lateral roots and shoot biomass (Figure 4).

The bacterial isolates exerted a significant influence on *Arabidopsis* growth characteristics (Figure 4 and Figure 5). Comparisons were made among 8 ARS-strains and a non-inoculated control. The relative increase in shoot and root biomass due to bacterial isolates ranged between 40 and 200%, over the un-inoculated control, whilst the corresponding increase in the root length and lateral root number ranged between 30–45% and 100–480%, respectively (Figure 5). In general, the effect of all multi-trait PGPR strains changed the root architecture (Figure 4 and Appendix A) dramatically. The efficacy of different isolates for growth characteristics was variable. Bacterial isolates 2SG20, 5SG3, 2C19, 2C23 and 3SG19 performed significantly better than others. Overall, the effect of bacterial inoculation was more pronounced on roots than shoots.

### 3.5. Plant Growth Promotion on A. thaliana (Col-0) under Salinity Conditions

We used the most salt tolerant *Bacillus* arylsulphatase-producing strains (1.SG.7, 5.SG.3) and two *Pseudomonas* strains (2.SG.20, 2.C.19). To determine their plant growth effect under gradient salinity of NaCl concentrations (37.5, 75, 150 and 200 mM), the four strains were co-cultivated with *A. thaliana* (Col-0) seedlings at 3 cm distance. The viability of *A. thaliana* (Col-0) plants under salinity conditions was very satisfactory in treatments with salt stress up to 75 mM, but from there on, it decreased dramatically, and while the spores germinated, they did not develop (Figure 6).

The results revealed that the four bacterial strains enhanced the number of lateral roots and shoot biomass. All multi-trait PGPR strains changed dramatically the root architecture under salt stress at concentrations 37.5 and 75 mM NaCl. Observing the control seedlings (without NaCl), we noticed that at concentrations of 37.5 and 75 mM NaCl, the plant fresh and dry weight significantly decreased, in contrast to the seedlings inoculated with the *Bacillus* strains 1SG7 and 5SG3 where root fresh and dry weight significantly increased (Figure 7), *Pseudomonas* strains 2SG20 and 2C19 kept the same weight level and increased the root length at 37.5 mM NaCl, while they decreased it with no significant difference at 75 mM NaCl (Appendix A). All inoculated seedlings presented a higher level of root length comparing to the control (Figure 7). Plant dry weight results followed the fresh weight ones. All seedlings, including control, developed more lateral roots under 37.5 mM NaCl stress.

### 3.6. Screening of 2 Selected Compatible Bacterial Strain Dyads as Mixtures for Plant Growth Promotion on A. thaliana (Col-0)

To investigate the growth-promoting ability of compatible bacterial strain mixtures (consortia), we tested 2 compatible dyads, i.e., the bacterial mixture Dyad A (1.SG.7 with 5.SG.3) identified as *Bacillus* sp. and *B. amyloliquefaciens*, respectively, and Dyad B (2.C.19 with 5.SG.3) identified as *P. moraviensis* and *B. amyloliquefaciens*, respectively. To determine their positive plant growth effect, the two dyad mixtures were co-cultivated with *A. thaliana* seedlings, inoculated at a distance. The results revealed that the two bacterial strain mixtures Dyad A and B enhanced the shoot and root biomass fresh and dry weight and also the root length and the number of lateral roots. Consortium Dyad B promotes plant growth more effectively comparing to the single strains (Figure 8).

### 3.7. Overall Additional Beneficial Traits of the Selected Strains

As a general assessment of the results, we compiled an overall of the phenotypic expression level of each of the additional beneficial characteristics of the selected arylsulphatase bacterial isolates by comparing the results based on the rating level of each trait among different strains (Table 3).

All assays were performed as described in Materials and Methods [36,37,38,39,40,41,42,43,44,45,46]. Table 3 was constructed according to the estimates of response level of each bacterial strain to the applied beneficial traits (strong, medium, low, not applicable). Rating scale is provided.

## 4. Discussion

The rhizospheric microbiome plays a crucial role in the growth and health of many plant species. The assembly of a plant root-associated microbiome is a dynamic, multistep process determined by dispersal, species interactions, the environment and the host [13,58,59,60,61].

Calcareous soils are commonly found in important agricultural areas, mainly around the Mediterranean, America and Australia [62]. Calcareous soils are problematic, usually poor in organic matter and available nitrogen and they require modification to promote their support for agriculture production. Calcareous soils in Greece are characterized by calcium carbonate from 20% to 80%, which makes them problematic for crop production development. The fertility of these soils has significant problems with the availability of nutrients due to the high pH > 7.0–8.3 [63]. A wide range of effective microorganisms can solubilize nutrients, and their colonization of roots promotes and enhances nutrient uptake [7,10,11,13]. The objective was to test these rhizospheric microorganisms already established on that poor problematic soil for their potential fertilizing and biocontrol ability. The selected strains could be used afterward as biofertilizers and/or biocontrol agents in such problematic soils and not only.

Previous studies of our group revealed that a core group of 68 rhizospheric ARS-producing bacteria mediated the unlocking of essential nutrients (Fe, P and S) from soils, thus enhancing the capability of wheat to take up nutrients from calcareous soil. As there are strong indications that plants select for traits rather than taxonomy [48], investigation on additional potential traits of this microbial community could provide valuable information on understanding the ‘hub microorganisms’ within the community. Such desirable characteristics should endow the bacteria with a high dispersal ability under harsh environmental conditions, good colonization ability, tolerance to adverse saline conditions and production of compounds that affect plant growth. Moreover, due to their coexistence, compatibility is an additional desirable trait along with antagonistic activity against phytopathogens.

The ARS-selected beneficial strains may survive in soil, when applied, for a longer period than other non-ARS producing strains under sulphate-limited conditions [18]. Microbial fertilization of some bacterial strains develops an alternative strategy dependent on arylsulfatase activity to fulfil their S requirements. It has been proven that an ARS microbial biofertilizer may improve or alter soil biological composition by affecting its fertility [48].

Biotic stress such as phytopathogen attacks caused by a plethora of fungi, bacteria and other microorganisms can be ameliorated by PGPR. Pseudomonas and Bacillus are predominant genera of PGPR with unique characteristics, diversity and relationship to plants, thus inducing sustainability [61,64,65]. In the present study, the bacterial strains at hand were tested for their antifungal activity against the phytopathogenic fungi *Rhizoctonia solani*, *Fusarium oxysporum*, *Botrytis cinerea* and *Colletotrihum* sp. in dual cultures. The reason for the selection of the aforementioned fungi is that all of them infect wheat, the crop that the rhizobacteria were isolated from, and a plethora of plant species as well. PGPB are used broadly as plant and soil inoculants as yield enhancements and biocontrol activators due to their eco-friendly attitude. Plant disease management using biocontrol functional biofertilizers could be a sustainable solution, as plenty of rhizospheric or endophytic bacterial strains may serve as biocontrol agents and plant growth promoters [66,67,68,69,70].

Indole acetic acid production of PGPR strains is one of their major properties, considered as one of the most physiologically active phytohormones that improves plant growth, supports plant cell division, influences the multiplication of root hairs, controls the differentiation of root meristem and affects the root system by increasing length and weight in main roots and lateral root number. These benefits enhance nutrients acquisition and improve plants’ development and yield [68,71,72,73]. The IAA producing quantity of the arylsulphatase-producing strains (ranged from 2.35 ± 0.6 to 28.35 ± 0.8 μM mL^−1^) seem to be effective for plant growth promotion of plants. IAA quantities like these seem to increase surface area and length of roots, loose cell wall and release exudates [74]. Extrinsic IAA in a developing plant controls various processes such as primary root elongation. A low-IAA producer strain, *Phyllobacterium brassicacearum* STM196, triggered changes in IAA distribution in *Arabidopsis* plant tissues, which was independent of IAA released by bacteria itself [75]. IAA production by microbes in certain amount can only promote host plant growth, omitting any negative effect at the pathogenic microbes which produce IAA, as bacteria use this phytohormone to interact with plants varying from pathogenesis to phytostimulation [76].

Swarming motility constitutes a key bacterial trait involved in many functions of plant-associated bacteria. One of these bacterial functions is root colonization ability. These modalities of motility may be harnessed for beneficial tasks through novel and ecologically safe strategies [77]. All 8 bacterial strains of this study presented enormous swarming motility in vitro at 30 and 37 °C, whilst the strains *P. fluorescens* 2C23, *Paenibacillus polymyxa* 5SG10, *Bacillus amyloliquefaciens* 5SG3 developed swarming colonies even at 42 °C. These results showed that the aforementioned strains could be remarkable root colonizers also at high-temperature environments.

The plant-associated biofilms protect plants from biotic and abiotic stress, flatten out the possible microbial competition and increase growth, yield, and crop quality [78]. Biofilm formation on plant roots by promising PGPR may be included as an additional criterion to select a better rhizosphere colonizer. Further, a study with mutant deficient in biofilm should be developed for comparative analysis to explore the exact contribution of biofilm in root colonization under the natural soil-plant system [79]. Researchers proved that biofilm-producing PGPR could be utilized as plant growth promoters, suppressors of plant pathogens, and alleviators of water-deficit stress at tomato plant pot experiments [80]. These selected strains could strengthen their role as excellent root colonizers that can be viable for long on the plant, even at high temperatures. The most important property of the successful colonization of the beneficial rhizospheric bacteria is the well adaptation to the rhizosphere in combination with their other beneficial traits [81]. The efficient colonization of the root surface is a strong advantage and the only chance for the bioinoculant to survive in a competitive and sometimes hostile environment. Thus, successful swarmers promote successful root colonization, even at high temperatures.

Plenty of studies substantiate that endophytic or epiphytic rhizospheric bacterial strains are functional and could effectively alleviate the toxicity of salt stress [40,82,83,84]. Salinity affects flowering and fruiting pattern and aberration in reproductive physiology, which ultimately influences crop yields and biomass [84].

PGPR that ameliorate salt stress in crop plants utilize an array of direct or indirect mechanisms producing auxins, gibberellins, cytokinins, synthesize ACC deaminase, secondary compounds such as exopolysaccharides and osmolytes (proline, trehalose and glycine betaines), regulate plant defence systems and activate plant’s antioxidative enzymes under salt stress [85,86,87,88,89,90,91,92,93]. Saline tolerant PGPR are potential enhancers of saline agro-ecosystems productivity [94], and this advantage makes them very important tools for farmers who want to keep the balance between production in such kind of soils and organic farming. Soil salinity inhibits plant growth, tissue development and affects all the activities involved with the rhizospheric bacteria [95]. It has been found that one of the most common bacterial genus often isolated from the roots of plants that survive and yield under salt stress is *Pseudomonas* sp., included *P. fluorescens*, *P. putida*, *P. stutzeri*, *P. mendocina* and *P. chlororaphis* [95,96].

*Pseudomonas* strains have successfully been tested to alleviate salinity effects on plants establishing root colonization, reducing toxic ions uptake, producing phytohormones, inducing systemic resistance and also inducing salt tolerance [97,98]. Bacillus strains also improve or induce salinity tolerance in different plant species [99,100].

The use of *A. thaliana* as a genetic model plant facilitated among others the elucidation of studies about salinity tolerance. *A. thaliana* wild type Col-0 is an easy and fast growth model plant that develops, reproduces and responds to stress and disease in much the same way as many crop plants. Although differences between Na^+^ accumulation and tolerance under salt stress may differ between *Arabidopsis* and cereals [100], the use of this model plant can certainly elucidate the possibility of inducing salinity in the presence of PGPR strains in crops.

In our study, both *Bacillus* (1.SG.7, 5.SG.3) and *Pseudomonas* (2.SG.20, 2.C.19) strains seem to mitigate salt stress in *Arabidopsis* plants in vitro at different NaCl concentrations (0, 37.5, 75, 150 and 200 mM). The fact that these two strains are resistant to salinity was almost to be expected since these strains have been isolated from rhizospheric soil of inferior quality, with natural irrigation and with an Olsen *p* value of 7.8 mg kg^−1^ [48]. The other six strains from the same rhizosoil showed relative tolerance and multi-trait benefits. *Arabidopsis* plant survival in salinity was excellent at 37.5 and 75 mM NaCl and much lower at 150 and 200 mM (Figure 6). Those treatments resulted in stunted plant growth throughout the experimental period. In general, *Arabidopsis* wild-type development is negatively affected by salt concentrations of more than 100 mM NaCl [101].

For a possible cumulative expression of their characteristics towards achieving a positive plant stimulation and protection, we must first ensure the sustainable coexistence of these strains, which means their compatibility. In the present study, the bacterial strains belonging to the genera *Bacillus* and *Pseudomonas* were tested as producers of antagonistic substances and as indicator strains to clarify if they antagonized each other. The 2C23 one was inhibited by 3 strains; also, the 2C19 and 2SG8 ones were mutually inhibited. Among them, seven could coexist since they did not show antagonistic effects on the growth of the examined bacterial strains. Therefore, we consider these eight strains with the potential of being part of specific bacterial mixtures. Although a single application can be effective, mixed inoculants as dyads or more strains together in one mixture could adapt to a broader range of environmental conditions and may possess various modes of action [12,15,102,103].

Thus, we tested one mixture, Dyad A, with two isolates belonging to the equal bacterial genus, *Bacillus* sp. and *B. amyloliquefaciens* (1.SG.7 with 5.SG.3), and another compatible mixture, Dyad B, with two isolates belonging to different bacterial genera together, *Pseudomonas moraviensis* 2.C.19 and *B. amyloliquefaciens* 5.SG.3. All three isolates (1.SG.7, 5.SG.3, 2.C.19) belong to the last selected multi-trait strains (Table 3). *B. amyloliquefaciens* strain 5.SG.3, which is a successful strain (Figure 5 and Figure 7) with biocontrol activity against phytopathogens, IAA production, salt tolerant, plant growth promotion under salinity etc. (Table 3) has been used in both bacterial mixtures. Our results indicated that both mixtures provided more impressive results concerning shoot and root biomass fresh and dry weight, root length per plant and lateral root number. Dyad B showed significantly different results in all the experiments we performed, not only comparing to the single strain application but also to Dyad A (Figure 8), indicating that Dyad B is a successful consortium for plant growth promotion. Microbes of a PGPM mixture in contact or in proximity or during the plant rhizosphere colonization competition assay are considered compatible when they have no growth suppressive effect on each other during their co-culture in vitro. Compatibility between strains may be achieved when one strain produces toxic compounds and the second strain possesses a detoxifying mechanism that could lead to a certain tolerance of the compounds and vice versa [12,104,105,106,107,108,109]. Antagonistic *P. fluorescens* mixtures against phytopathogens have failed to inoculate plants in contrast to single strains due to their incompatibility [110,111]. The mixture of *Pseudomonas* strains WCS417r and SS101 had a contentious result due to their semi-incompatibility. The density of WCS417r fluctuated against *Pf*. SS101 resulted in minor inoculum antagonism [112]. In parallel, *Bacillus*-based mixtures, which are widespread, have similar particularity as *Pseudomonas* mixtures, meaning that bacillus mixtures occasionally have synergistic or antagonistic functions that may lead to failed inoculums. This phenomenon has already been observed years ago [14,113], leading to timely compatibility test for successful mixture construction. Moreover, these strains can form viable mixtures, so they can be used as sustainable biofertilizers. The fact that these strains can retain the ability to promote plant growth under salinity conditions gives an additional advantage to these strains, and they could potentially be developed as inoculants to alleviate the salinity stress in the plants grown in arid or saline soils. In a recent study, researchers validated the potential of *P. aeruginosa* strain FB2 and *B. subtilis* strain RMB5 as biofertilizer and biopesticide agents and proved their potential as antagonistic plant-beneficial bacteria effective against a range of fungal phytopathogens. Both of these bacteria can be used to develop a dual-purpose bacterial inoculum as biopesticide and biofertilizer [114].

It is important that this study constitutes an evaluation of selected strains of Arylsulfatase-producing rhizobacteria based on the combination of remarkable characteristics. The compatible bacterial group of four bacterial strains is a potential active consortium consisting of four compatible members, resistant, competitive against phytopathogens, with plant growth promoting abilities, good colonizers, all-temperature resistant with one member highly resistant and also salt tolerant, promoting plant growth under salinity conditions. In a recent study, researchers proved evidence that resistant bacteria respond by activating some inherent mechanism to resist an antagonistic substance [115]. Other researchers have studied the metabolic cost for the resistance and/or antagonistic strains to produce their active substances and the effect of their growth [116,117]. Resistant, less resistant and antagonistic strains together in a consortium can play the rock-paper-scissors game [117,118]. Thus, in future experiments, our research team will test the effectiveness of the four-member consortium in combinations of two or more strain mixtures not only with in vitro but with in vivo experiments in wheat plants under normal and salinity conditions, studying their biofilm formation and effect in colonization and growth. Additionally, we will continue our research with genome mining approaches, showing the antimicrobial biosynthetic gene clusters of the selected four multi-trait strains from calcareous soil as a functional tool in plant growth promotion, and biological control studies leading to eco-friendly plant protection.

## 5. Conclusions

In this study, we further screened the arylsulfatase (ARS)-producing bacteria for beneficial traits, assessed their growth promotion activity in vitro, proved their ability to form biofilms and evaluated their resistance under salinity stress in vitro and the effectiveness of selected salinity tolerance. A selected group of arylsulfatase (ARS)-producing bacterial isolates with biocontrol activity and resistance to salinity stress can stimulate seed germination and enhance plant growth under salinity stress conditions. Moreover, we evaluated compatible mixtures of these multi-trait isolates for their ability on plant growth promotion, proving that a mixture consisting of one *Pseudomonas* and one *Bacillus* strain could give remarkable results compared to single strain application. 

The multi-trait rhizobacteria, which are active constituents of the arylsulphatase producing rhizobacteria community, can improve, promote and potentiate the natural beneficial microbe-host interactions in agro-ecosystems. Generally, the use of the appropriate compatible biofertilizers with beneficial characteristics concerning growth promoting and biocontrol activity in combination with disease prediction systems can prevent and solve serious crop disabilities and aim at a sustainable agriculture.

## Figures and Tables

**Figure 1 microorganisms-09-01588-f001:**
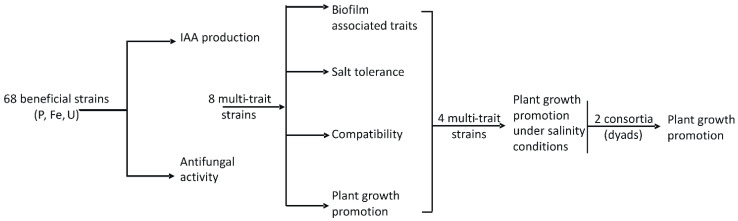
Experimental strategy plan. Bacterial strain selection steps.

**Figure 2 microorganisms-09-01588-f002:**
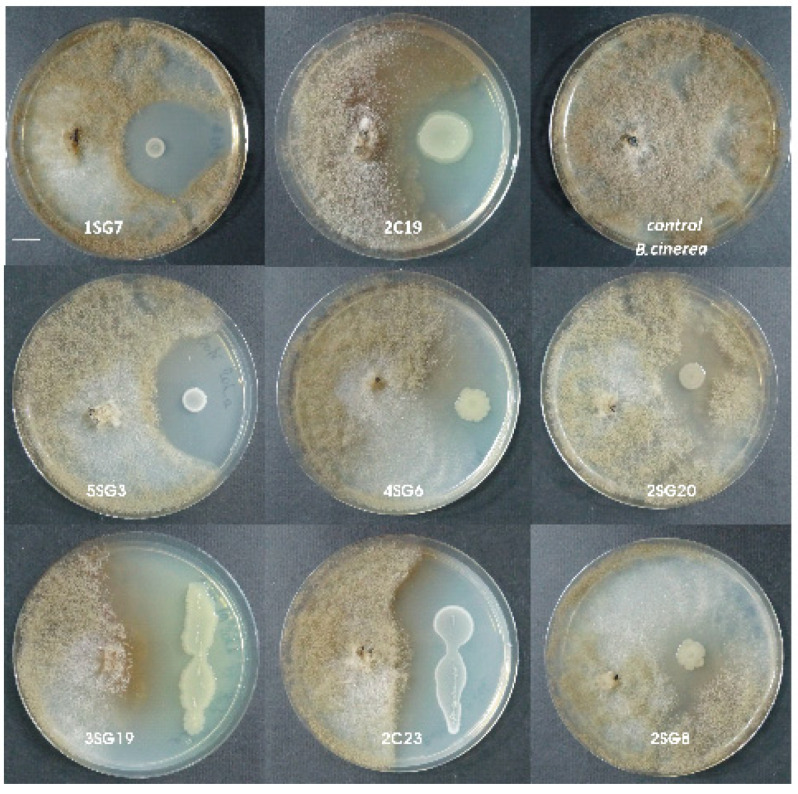
Antagonistic activity of the eight selected bacterial strains against *Botrytis cinerea*. Scale bar equals 1 cm.

**Figure 3 microorganisms-09-01588-f003:**
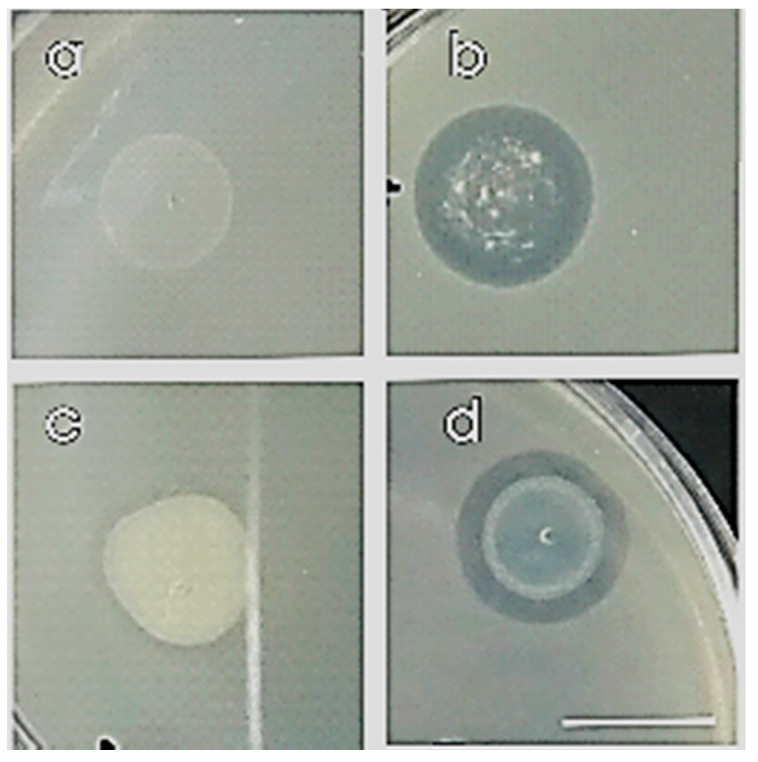
Compatible (**a**,**c**) and non-compatible (**b**,**d**) strains on co-cultured NA petri dish with overlay method after 24 h incubation at 30 °C. Scale bar equals 1 cm.

**Figure 4 microorganisms-09-01588-f004:**
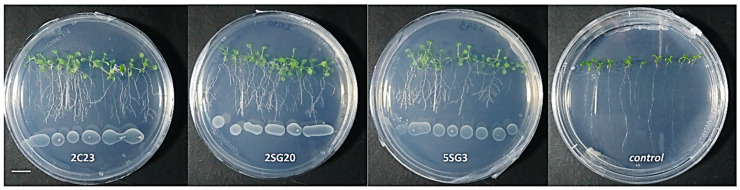
Representative phenotypic responses of co-cultivation of *A. thaliana* seedlings with selected bacterial strains inoculated at 3 cm distance from root tips. Scale bar equals 1 cm.

**Figure 5 microorganisms-09-01588-f005:**
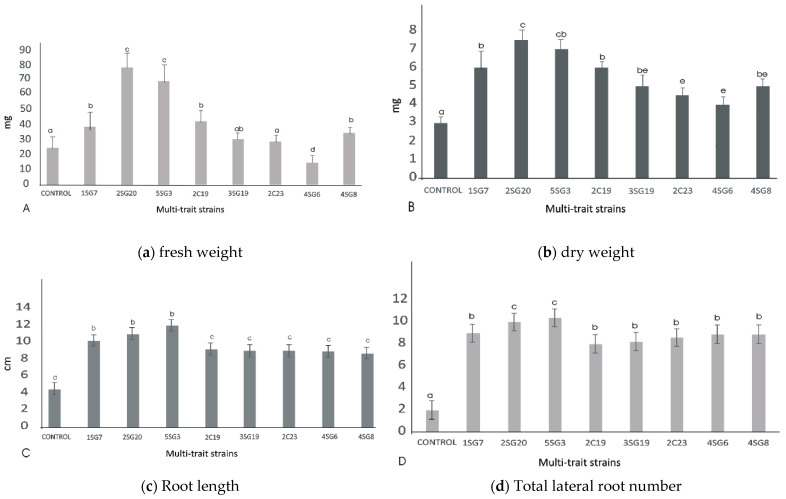
Effect of the eight multi-trait PGPR inoculation on growth performance of *Arabidopsis* seedlings (grown in MS agar plates). (**a**) Fresh weight per plant (shoot and root biomass). (**b**) dry weight per plant (shoot and root biomass). (**c**) Main and secondary root length per plant. (**d**) Total lateral root number per plant. Data points are the mean values of triplicate assays while bars indicate ±SD. The error bars represent the least significant difference among treatments at *p* ≤ 0.05. Different letters indicate statistically significant difference (*p* < 0.05).

**Figure 6 microorganisms-09-01588-f006:**
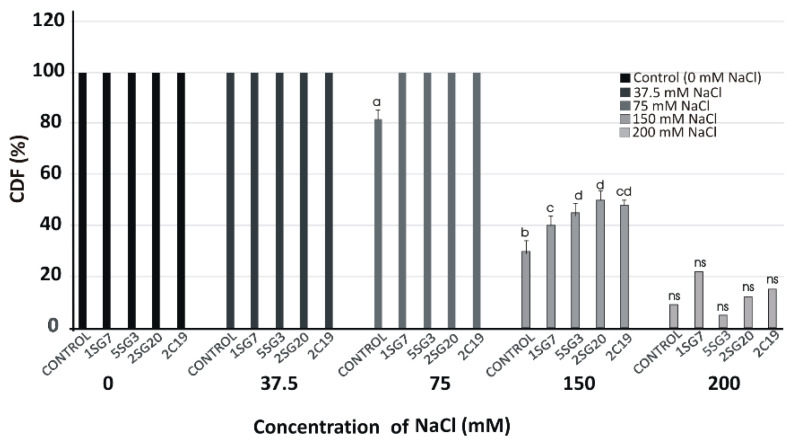
Viability of *A. thaliana* (Col-0) plants on different NaCl concentrations (37.5, 75, 150 and 200 mM) in the presence of the four most salt tolerant strains (1.SG.7, 5.SG.3, 2.SG.20, 2.C.19), recorded as a percentage of seedlings having green cotyledons over the total number of seeds germinated (Cotyledon Development Frequency, CDF) on half-strength MS medium for 15 days. Data points are the mean values of triplicate assays while bars indicate ±SD. The statistical significance level is indicated by different letters on top of each bar. *p* < 0.05 compared to the control; ns, non-significant.

**Figure 7 microorganisms-09-01588-f007:**
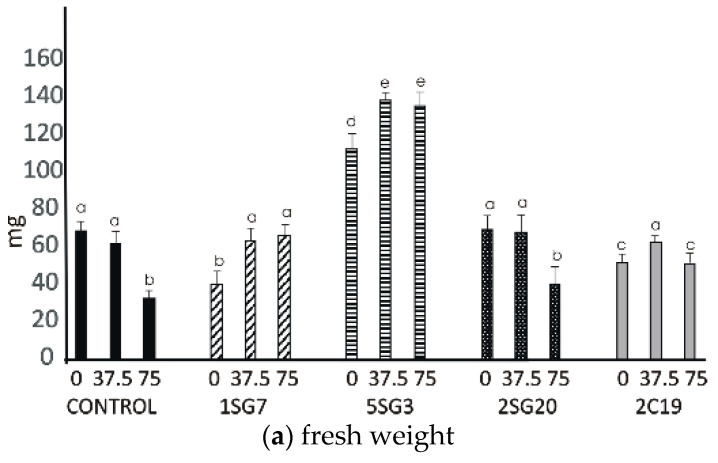
Effect of salt tolerant PGPR inoculation on growth performance of *Arabidopsis* seedlings under salinity conditions (0, 37.5, 75 mM NaCl): (**a**) shoot and root biomass fresh weight of *A. thaliana* grown in MS agar plates, (**b**) shoot and root biomass dry weight of *A. thaliana* grown in MS agar plates, (**c**) root length per plant. The error bars represent the least significant difference among treatments at *p* ≤ 0.05. Different letters indicate a statistically significant difference (*p* < 0.05).

**Figure 8 microorganisms-09-01588-f008:**
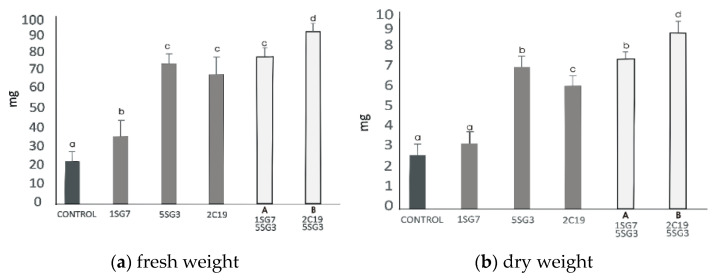
Effect of bacterial mixtures in dyads Dyad A (1.SG.7 with 5.SG.3) and Dyad B (2.C.19 with 5.SG.3): (**a**) shoot and root biomass fresh weight of *A. thaliana* grown in MS agar plates, (**b**) shoot and root biomass dry weight of *A. thaliana* grown in MS agar plates, (**c**) root length per plant. The error bars represent the least significant difference among treatments at *p* ≤ 0.05. Different letters indicate a statistically significant difference (*p* < 0.05).

**Table 1 microorganisms-09-01588-t001:** Fist selection screening of eight rhizospheric arylsulfatase (ARS)-producing bacterial isolates based on IAA production and antifungal activity.

Isolate	P Solubilization	Siderophore	Urease Activity	Identification Based on 16S rRNA	Accession Number	Antifungal Activity Inhibition Rate (IR)	IAA Production μg/mL
1.SG.7	-	+	+	*Bacillus* sp.	LR027398	56.5	2.35 ± 0.6
2.SG.20	+	+	+	*P. koreensis*	LR027423	50.2	18.25 ± 0.9
5.SG.3	+	+	+	*B. amyloliquefaciens*	LR027456	62.9	28.35 ± 0.8
2.C.19	+	+	+	*P. moraviensis*	LR027411	56.5	25.15 ± 0.5
3.SG.19	+	+	+	*P. fluorescens*	LR027436	53.9	15.41 ± 0.2
2.C.23	+	+	+	*P. fluorescens*	LR027412	55.6	22.61 ± 0.8
4.SG.6	+	+	+	*P. koreensis*	LR027448	54.5	20.16 ± 0.9
2.SG.8	+	+	+	*P. koreensis*	LR027414	52.8	11.2 ± 0.9
[48]						present study	

P phosphate solubilization experiment was performed according to Pikovskaya, 1948 [50]; siderophore production was performed according to Schwyn and Neilands, 1987 [51], urease activity according to Bouranis et al., 2019 [48]; IAA production was estimated according to Gordon and Weber, 1951 [52], Bric 1991 [53]; antifunagal activity was estimated as remarkable when IR exceeded 50% in our in vitro experiments for the phytopathogens that we tested (numbers are means of IR value for the phytopathogenic fungi that we tested).

**Table 2 microorganisms-09-01588-t002:** Strain compatibility assay using overlay method.

Isolate	1.SG.7	2.SG.20	5.SG.3	2.C.19	3.SG.19	2.C.23	4.SG.6	2.SG.8
1.SG.7	+	+	+	+	+	-	+	+
2.SG.20	+	+	+	+	+	-	+	+
5.SG.3	+	+	+	+	+	-	+	+
2.C.19	+	+	+	+	+	+	+	+
3.SG.19	+	+	+	+	+	+	+	-
2.C.23	-	-	-	+	+	+	+	+
4.SG.6	+	+	+	+	+	+	+	+
2.SG.8	+	+	+	-	+	+	+	+

The compatibility assay was performed in vitro among the 8 selected potential PGP strains using the overlay method using drop technic [43]. Compatible dyads were detected by the formation of colony growth (Figure 3a) and non-compatible by the formation of transparency (Figure 3b) where the drop was placed after at least 24 h (Compatible +, Non compatible -).

**Table 3 microorganisms-09-01588-t003:** The overall Plant Growth Promoting, biofilm associated traits and compatibility skills of 8 multi-trait bacterial strains, in brief.

Isolate	Biocontrol Activity	IAA Production	Plant Growth Promotion	Plant Growth Promotion in Salinity	Lateral Roots	Strain Salinity Tolerance	Temperature Tolerance	Biofilm Formation	Compatibility in Dyads
1.SG.7									
2.SG.20									
5.SG.3									
2.C.19									
3.SG.19									
2.C.23									
4.SG.6									
2.SG.8									
	strong								
	medium								
	low								
	not applicable								

## Data Availability

All the bacteria strain data used in this study are available in the NCBI database at Bioproject: PRJEB28499.

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
