# Peer review of "Multi-Trait Wheat Rhizobacteria from Calcareous Soil with Biocontrol Activity Promote Plant Growth and Mitigate Salinity Stress"

_microorganisms, 2021, doi:10.3390/microorganisms9081588_

Round 1

Reviewer 1 Report

The manuscript entitled “Multi-trait wheat rhizobacteria from calcareous soil with biocontrol activity promote plant growth and mitigate salinity stress” by Venieraki et al. is interesting and worth publishing.

Suggestions and corrections

  • The abstracts of this paper start with an introduction to the PGPR application. It should provide current work and results.
  • Line 35-38 sentences are not clear. Should be rewritten.
  • The introduction part has a piece of minimal information about ARS (arylsulfatase). The authors should add more valuable information.
  • The author has not explained the selection of calcareous soil in this study.
  • The authors should comment about why Arabidopsis thaliana Col-O was taken for this study.
  • For the antifungal activity, the authors should provide the figure of the eight strains.
  • The authors should provide a clear image of the compatible analysis results.
  • Line 372- 30oC should change to 30 °C.
  • The authors should clearly explain the dyad of bacterial culture used in this study.
  • Table 3 - why you have not performed “Heat-Map” analysis. It will help to provide clear results.
  • The authors should provide clear images of supplementary (Figure S1 and S2) images.
  • The conclusion part is very poor. The authors should clearly summarize the main finding of the research.
  • Overall the manuscript is satisfactory, well written. However, I would suggest thorough proofreading for rectifying grammatical and usage errors if any.

The manuscript can be considered for publication, only after satisfying all these queries.

Reviewer 2 Report

Authors modified the manuscript quite well but still discussion section needs more attention. 

  1. Why Arylsulphatase activity with microbes is done, there is no discussion about it.
  2. Scientific name should be in italic, correct it throught out the manuscript.
  3. In Table 1, instead of Fe, write Siderophore directly.

Round 2

Reviewer 1 Report

No further comments.